# Monitoring the Health of Coastal Environments in the Pacific Region—A Review

**DOI:** 10.3390/toxics11030277

**Published:** 2023-03-18

**Authors:** Louis A. Tremblay, Anthony A. Chariton, Meng-Shuo Li, Yong Zhang, Toshihiro Horiguchi, Joanne I. Ellis

**Affiliations:** 1Cawthron Institute, Private Bag 2, Nelson 7042, New Zealand; 2School of Biological Sciences, University of Auckland, Auckland 1142, New Zealand; 3Department of Biological Sciences, Macquarie University, Sydney, NSW 2109, Australia; 4State Key Laboratory of Marine Environmental Science of China, College of the Environment and Ecology, Xiamen University, Xiamen 361102, China; 5Ecosystem Impact Research Section, Health and Environmental Risk Division, National Institute for Environmental Studies, 16-2, Onogawa, Tsukuba 305-8506, Ibaraki, Japan; 6School of Sciences, Waikato University, Tauranga 3240, New Zealand

**Keywords:** China, Japan, New Zealand, Australia, environmental management, anthropogenic stressors, multiple stressors, climate change, indigenous knowledge

## Abstract

Coastal areas provide important ecological services to populations accessing, for example, tourism services, fisheries, minerals and petroleum. Coastal zones worldwide are exposed to multiple stressors that threaten the sustainability of receiving environments. Assessing the health of these valuable ecosystems remains a top priority for environmental managers to ensure the key stressor sources are identified and their impacts minimized. The objective of this review was to provide an overview of current coastal environmental monitoring frameworks in the Asia-Pacific region. This large geographical area includes many countries with a range of climate types, population densities and land uses. Traditionally, environmental monitoring frameworks have been based on chemical criteria set against guideline threshold levels. However, regulatory organizations are increasingly promoting the incorporation of biological effects-based data in their decision-making processes. Using a range of examples drawn from across the region, we provide a synthesis of the major approaches currently being applied to examine coastal health in China, Japan, Australia and New Zealand. In addition, we discuss some of the challenges and investigate potential solutions for improving traditional lines of evidence, including the coordination of regional monitoring programs, the implementation of ecosystem-based management and the inclusion of indigenous knowledge and participatory processes in decision-making.

## 1. Introduction

The consequences of current anthropogenic activities are resulting in impacts that extend beyond the safe operating space of the planetary boundary based on a weight-of-evidence framework [1]. Coastal watersheds that support more than one half of the world’s human population are particularly affected, as they experience unprecedented urban, agricultural and industrial expansion [2]. Analyses of ecological change indicate that human activities lead to impacts on marine environments, and a large proportion of impacts are strongly affected by multiple anthropogenic drivers [3,4]. Research has shown that the consequences of human activities have pushed estuarine and coastal areas far from their historical baseline of diverse and productive ecosystems [5]. The impacts on coastal sediment quality are not exclusively an environmental issue, as they also directly affect human activities such as tourism, fisheries, aquaculture and recreation. Moreover, the impacts can lead to costly remediation exercises, including navigational dredging and environmental restoration projects [6]. Pollutants from land-use/anthropogenic activities have widespread effects on coastal and freshwater ecosystems [7]. Similarly, pressures from these activities can lead to both chronic and point-source pollution, which can result in cumulative impacts on the health of marine ecosystems [3,8].

Anthropogenic pressures, including nutrient loading, run-off of pollutants and sediments, fishing pressures, invasive species, habitat loss and climate change impact biodiversity in vulnerable coastal habitats such as coral reefs, which are one of the most biologically diverse and productive systems on the planet [9]. Estuarine invertebrate communities are often threatened by multiple stressors, including eutrophication, industrial pollutants, dredging, shoreline development, invasive species and overfishing [10]. There is still uncertainty regarding the responses of invertebrates to climate change and their interactions with other stressors, including the possible synergistic or antagonistic effects dependent on the combination of stressors [4,11]. Pollution also impacts most marine protected areas (MPAs), which have been established to protect habitats, biodiversity and ecological processes as well as helping to maintain fishery productivity [12]. Conservation protection initiatives such as MPAs will not fully address the ongoing loss of marine and terrestrial biodiversity. To maintain human and ecosystem health, alternative solutions based on sustainable targets for human demand for ecological goods and services must be part of the international conservation agenda [13].

Many of the marine hotspots that are linked to the most rapid ocean warming occur where human dependence on marine resources is greatest, particularly on the East and Southeast Asian coasts; therefore, these areas are of critical concern in the context of food security [14]. Potential changes in marine resource distribution and abundance are significant in these densely populated tropical regions of high biodiversity, where small-scale fishing operations constitute more than 90% of the world’s fish traders. For example, the expanding Chinese marine fisheries are now suffering from the depletion of resources due to multiple anthropogenic threats, including overfishing, pollution and land reclamation [15].

In the context of multiple stressors, it is important to characterize and assess the extent to which pollutants impact receiving ecosystems. The identification of the causative stressor(s) responsible for the impacts is required and important for informing the development of marine policy [16,17]. Risk assessment of a marine environment is conducted by measuring pollutants and the effects on receptor species in the targeted ecosystem [18]. The conflict between the ongoing drive to achieve greater economic returns from the oceans and the resulting degradation of many marine ecosystems prioritizes the management of the effects of multiple stressors [3]. The pressure on marine ecosystem services is likely to increase with the world’s rapidly growing human population. To address this challenge, legislative initiatives have been developed, including the European Union Marine Strategy Framework Directive (MSFD), which is a good example of regulation that aims to have marine waters achieve Good Environmental Status (GES) [19]. Another initiative is the marine integrated contaminant monitoring (ICON) program under the Oslo and Paris Conventions (OSPAR), which has adopted an approach based on the integration of chemical and biological methodologies to assess the impacts of contaminants in northern European coastal areas [16].

Globally, there is a need for integrated approaches that consider the close relationships between the collection, processing and analysis of marine water quality data sets and the decision-making processes linked to policy-making [17]. In this preferred framework, the use of integrated methods combining economic, social, environmental and ecological priorities, such as multiple criteria analysis methods, can help all parties to rationalize their views and be informed by the best available science [17]. 

Coastal and ocean ecosystems provide commercial, cultural, recreational and economic benefits as well as support diverse habitats and species of local and global significance. Reviews of environmental issues in the Pacific [20] and a synthesis report on the key threats to the Pacific Ocean [21] have identified pollution, habitat destruction, overfishing and exploitation, climate change and invasive species as the five most critical threats to the ocean’s sustainability and health. However, despite these environmental pressures, a recent study suggested that in comparison to other regions of the world, “the Pacific is relatively healthy” [22]. The objective of this review is to summarize the current monitoring frameworks in four Asia-Pacific regions and identify research gaps relating to the need to sustainably manage the health of coastal zones.

## 2. Main Asia-Pacific Regions

The large Asia-Pacific region has diverse ethnicities, population densities and climate types. There are also wide variations in the levels of human activities and impacts on ocean ecosystems. For example, Japanese waters have been found to be highly impacted, while the Torres Strait in northern Australia is one of the least impacted regions globally [3]. In the South China Sea, the ranking of the effects of coastal contaminants indicated that deterioration in water quality was the major concern, followed by biological impacts, which are less well demonstrated, and third, contamination of sediments [23]. Sedimentation is one of the major global threats to reefs, particularly in the Caribbean, Indian Ocean and South and Southeast Asia [8]. Some highly developed industrial areas of the Asia-Pacific region are facing frequent formations of oxygen depleted zones, as seen in other parts of the world [24]. South and East Asia have some of the highest rates of riverine transport of dissolved inorganic nitrogen to estuaries, which is derived from a range of sources including industrial development and the discharge of untreated wastes and effluents. The ongoing degradation in the East Asian seas is threatening global marine biodiversity and the functional integrity of key marine ecosystems such as coral reefs [25]. Several reports are available on the extent of water pollution and the consequences in South Asian countries. 

Countries in Oceania, a diverse region encompassing Australia, Melanesia, Micronesia, New Zealand and Polynesia, have a high level of species extinction due to increasing human populations and intensification of agricultural activities [26]. Another challenge facing South Asian and Pacific countries within the low-elevation coastal zone (LECZ) is the high vulnerability of these populations to the impacts of climate change, including sea-level rise [27]. In Australia and New Zealand, the pressure of chemical contaminants on the health of ecosystems is important and has been identified as a priority research focus in the Australasian component of the Society of Environmental Toxicology & Chemistry (SETAC) Global Horizon Scanning Project, which identified the question: “How can we identify and prioritize contaminants (traditional and emerging stressors) for sustainable management of ecosystems within different biogeographic regions?” [28]. The following sections review the range of pressures, the current legislative frameworks and the monitoring practices in four countries of the Asia-Pacific region.

### 2.1. New Zealand 

New Zealand is responsible for managing a large marine estate relative to its population size, with New Zealand’s Exclusive Economic Zone (EEZ) being the fourth largest in the world. The marine estate contains a wealth of natural biodiversity and supports marine tourism, recreation, oil and gas production, fisheries and aquaculture, with the potential for future economic opportunities, including renewable energy and seabed minerals.

An expert assessment identified the top threats to New Zealand marine ecosystems as ocean acidification and rising sea levels and temperatures, which are the result of global climate change, followed by land-based sedimentation and bottom trawling [29]. However, other dominant pressures identified include coastal development, land reclamation, marine pollution, aquaculture, wild and recreational fisheries, dredging and offshore mining. In this review, we focus on the top anthropogenic pressures identified by the New Zealand Ministry for Primary Industries (MPI); these pressures predominantly originate from climate change, land-based activities and fishing pressures.

Climate change can affect ocean acidification, sea levels and surface temperatures altering currents, stratification and mixing. In addition, there is potential for increasing frequency and intensity of storms [30,31]. In a New Zealand context, acidification was identified as a top threat to marine habitats because of changes in the ocean’s pH due to the uptake of carbon dioxide from the atmosphere. The primary concern relates to declines in ocean pH associated with corresponding reductions in carbonate ions, which can affect calcifying species and their ability to produce calcium carbonate structures such as shells [32]. Acidification can also influence growth, survival and reproduction of non-calcifying organisms [33,34]. Changes in ocean acidification therefore can have direct impacts on fisheries and aquaculture and, more widely, on marine food webs, key species and habitats.

Land-based activities related to urbanization and modification of surrounding catchments can affect coastal and marine environments through, for example, increased sedimentation and nutrient run-off, wastewater discharges, coastal reclamation and metal contamination from stormwater overflows. In New Zealand, land-based sedimentation was ranked as the third equal highest threat to the marine environment [29]. Historical and ongoing changes in New Zealand’s catchment land use due to large-scale clearances of forests and the expansion of livestock farming and forestry have resulted in increased terrigenous sediments and nutrient levels being delivered to coastal environments. Sedimentation has direct and indirect effects on marine ecosystems [35]. Direct impacts include clogging of the gills of filter feeders, decreases in filtering efficiencies [36], the smothering of shellfish beds and reductions in foraging abilities of aquatic animals [37]. Indirect effects include modifications to important marine habitats, such as nursery areas with biogenic species such as seagrass meadows, sponge gardens, bryozoan mounds or mussel and oyster reefs.

Alongside sedimentation, bottom trawling was ranked as the third highest overall threat to New Zealand marine habitats [29]. Contact fishing gear has been shown to homogenize soft sediment habitats, alter benthic assemblages and reduce marine biodiversity [38]. New Zealand’s coastal environments support highly valuable finfish and invertebrate fisheries, with most now considered to be fully exploited [39]. Many of these fisheries have a history of heavy exploitation in their initial phases [39], resulting in the introduction of a quota management system [40]. Of particular concern are New Zealand’s deep-sea fisheries (including orange roughy, oreros and rattails) that are highly vulnerable to overfishing because of slow growth rates and thus limited population recovery. Habitat disturbance by bottom trawling and dredging, which involves dragging a net or dredge along the seafloor, represents a significant threat to marine benthic biodiversity [41]. The loss of important bioturbators due to dredging also has implications for nutrient cycling in benthic habitats [42].

Overall, in New Zealand, 65 stressors were identified, based on accumulated expert opinion, and ranked by the severity of their likely impact and the number of habitats they could potentially affect [29]. The identification of these stressors is now being used to assist with the implementation of monitoring frameworks. Notably, considerations in selecting monitoring indicators highlight the use of the pressure–state–response frameworks for environmental monitoring. 

#### 2.1.1. Regulations and Guidelines

New Zealand has two main pieces of legislation for marine waters. In coastal waters (from the mean high water springs line out to 12 nautical miles), environmental effects are managed under the Resource Management Act 1991 (the RMA) [43]. Within this zone, regional councils are responsible for managing environmental effects in the coastal marine areas as well as on land. Notably, State of the Environment (SOE) monitoring is implemented by local regional councils as part of RMA obligations. The RMA requires that councils promote the sustainable management of natural and physical resources in areas where SOE monitoring and reporting can help determine whether these requirements are being met. SOE monitoring also helps with policy development and informs decision-makers of the consequences of actions and changes in the environment. Beyond 12 nautical miles and out to the extended continental shelf boundary, including the EEZ, environmental effects are managed under the EEZ Act [44], for which the New Zealand Environmental Protection Authority (NZ EPA) is the responsible agency. The purpose of the EEZ Act is to promote sustainable management of natural resources. The EEZ Act defines sustainable management in a similar way to the RMA and aims to balance the management of natural resources for economic benefit with the needs of future generations. In other words, the EEZ Act seeks to preserve the life supporting capacity of the environment and avoid, remediate or mitigate the adverse effects of activities on the environment.

#### 2.1.2. Monitoring

There are relevant monitoring frameworks in New Zealand that facilitate sustainable management [45]. The MPI monitors catch levels and the relative abundance of commercial fish stocks under the Fisheries Act 1996. The Act also requires ongoing assessments of the broader impacts of fishing and clearly outlines the need to “avoid, remedy, or mitigate any adverse effects of fishing on the aquatic environment”. In addition, the Act states that the “biological diversity of the aquatic environment should be maintained”. New Zealand also has a fully developed Marine High Risk Site Surveillance program to monitor non-indigenous species at harbours that are the first entry points for international vessels.

The Department of Conversation (DOC) is responsible for monitoring the marine environment and has identified the concept of ecological integrity as the basis on which to assess the state of New Zealand’s waters [45]. Ecological integrity is based on the assessment of four themes: nativeness, pristineness, diversity and resilience. DOC has also reviewed the monitoring programs conducted in marine reserves to enable national reporting of the status and trends occurring in these environments [45].

While there are many pieces of legislation and responsible authorities, New Zealand is recognized internationally for its environmental management and innovative regulatory frameworks, as demonstrated, for example, by the implementation of the first no-take marine reserve in 1975 [46] and the introduction of a quota management system for fisheries in the 1980s [47]. Māori, the indigenous people of New Zealand, have specific rights as parties of the Treaty of Waitangi [48], the foundation document of modern New Zealand. A major element of environmental management in New Zealand includes the recognition in policies and regulations of Māori connections with the oceans [44]. There is a strong commitment to wise stewardship of natural resources that includes close cultural connections to New Zealand’s waters [49].

### 2.2. Australia 

The vast majority of Australia’s 22 million people live in five coastal cities, with 85% of the population being within 50 km of the coast [50]. Australia has the third largest marine jurisdiction in the world, encompassing 13.86 million km^2^ [51]. Given its size, mainland Australian coastal waters capture a diverse range of environments, from temperate southern waters to tropical reefs and floodplains in the north. The marine environment contributes approximately AUD 50 billion per annum to the country’s economy and is expected to double within the next five years [52]. Tourism is also a significant contributor to the Australian economy, accounting for approximately 3% of the country’s GDP [53]. In particular, tourism associated with the Great Barrier Reef (GBR) makes up approximately 11% (AU$6.4 billion) of Australia’s domestic and international tourism [52].

As the world’s oldest surviving indigenous culture, First Nations Australians have a long and strong connection with coastal environments The resources associated with coastal systems are vital for activities such as hunting and fishing, and the coastal environments also contain numerous sites of cultural significance. Given the vulnerability of coastal systems to climate change-induced sea-level rise, the implications of sea level are likely to be profound for many Australian communities. In the World Heritage Kakadu National Park, Dutra et al. [54] highlighted the complex implications of sea-level rise on the park’s indigenous communities, including the direct and negative impacts on hunting and fishing and access to sacred sites, and the economic implications due to predicted declines in tourism. To date, the realization of the predicted implications of sea-level rise have focused on the loss of amenities around the coastal cities [55], despite the knowledge that regions such as Kakadu’s world-famous wetlands, including sites of indigenous cultural significance, will be largely lost to sea-level rise by the beginning of the next century [56]. In many regions, the ecological, social and economic impacts of climate change are already evident. Notable examples are the two mass bleaching events in the GBR, with the most recent occurring in 2016 [57]. Clearly, the implications of sea-level rise extend far beyond a decline in the ecological integrity of the GBR and cascade as losses in ecological services, tourism and social connections to the landscape. Other large-scale die-offs have recently been observed in coastal systems, including hundreds of kms of mangrove forests.

While climate change is a key factor contributing to the loss of the GBR’s ecological integrity, a number of other factors are also significant. One of the most notable is the agricultural run-off from the adjacent Queensland mainland, which produces increased loading of fine sediment as well as pulses of nutrients and pesticides. Herbicides, such as diuron, not only impact their targeted taxa but are also toxic to other autotrophs, such as the symbiotic algae associated with corals. While still contentious, there is considerable evidence to show that increases in nutrients are driving population booms in the Crown-of-Thorns starfish, a veracious predator of corals. As well as clogging corals with fine sediment, the collective effects of agricultural run-off have the capacity to reduce the GBR’s overall ecological condition, making it more susceptible to climate change. Numerous significant on-ground programs are in place to reduce run-off into the reef; however, it is difficult to balance the needs of the multiple users of the GBR environment. While the Queensland Government undertakes regular water quality monitoring of the region, no routine ecological monitoring has been performed to date. Given the size and diversity of the system, routine monitoring using traditional approaches (e.g., macrobenthic surveys) is unfeasible; however, new opportunities exist as a result of the development of molecular tools such as eDNA metabarcoding [58,59,60].

The impact of contaminants is by no means restricted to the GBR. For example, Sydney, like most other large coastal cities, has a marked influence on its surrounding coastal environments. A current issue is the management of run-off from stormwater, which can contain a complex mixture of industrial chemicals, pharmaceuticals, plastics and nutrients. Notably, coastal water quality in Sydney Harbour has dramatically increased in the last few decades. It is likely that this improvement is the combined result of the transition to depositing treated wastewaters offshore and the introduction of more articulate environmental protection laws, regulations and enforcement. However, current chemical stressors remain a concern, and much of Sydney Harbour is still significantly impacted by legacy metals and organic contaminants [61]. Most notable are the high sediment concentrations of dioxins in the Parramatta River (the western estuary of Sydney Harbour), where fishing is still banned almost four decades after the operations of chemical polluting companies ceased [62]. Similarly, metal contractions remain high in many regions of the harbour. Despite the significant recent efforts to remediate the harbour environment, numerous developments within the surrounding area are likely to oxide the contaminated sediments and thus increase their bioavailability and the risks to biota.

#### 2.2.1. Regulations and Guidelines

The key piece of legislation to protect Australia’s biodiversity and encompass coastal and marine environments is the Environment Protection and Biodiversity Conservation Act 1999 (EPBC Act). The EPBC Act aims to protect areas of significance, both ecologically and culturally, and promote sustainable use. While the Act is pertinent to world heritage areas, including the GBR, it is a federal law and pertains to areas of federal control (e.g., offshore waters beyond state jurisdictions, with states and territories responsible for their own matters). Consequently, for the most part, the Act does not apply to mainland coastal waters, as these areas are legislated by specific states and territories. While Australia’s regulations generally operate at a regional level, an exception is the oil and gas sector. In 2012, the National Offshore Petroleum Safety and Environmental Management Authority (NOPSEMA) was established in response to an oil spill inquiry (The Montara). As well as monitoring the industry’s safety and well integrity, NOPSEMA regulates the environmental management of the oil and gas sector at a national level.

One of the most important initiatives for the protection of coastal waters is the Australian and New Zealand Water Quality Guidelines (ANZWQG) [63]. The ANZWQG is founded on the following key elements: (1) community values; (2) conceptual models; (3) guideline values; (4) monitoring; (5) stakeholder involvement; (6) weight of evidence; and (7) location. Of particular note is the weight-of-evidence element, which uses multiple sources of evidence as part of the decision-making process. The sources of evidence include pressures, which are external activities that affect water quality; physical, chemical and non-water quality stressors (e.g., flow and catchment alterations); and ecosystem receptors, i.e., biodiversity, toxicity and biomarkers. Importantly, the ANZWQG is continually evolving and has a particular focus on supporting the improvement of toxicity data for many known key contaminants. However, it is emphasized that the ANZWQG are guidelines only and thus polluters may not face criminal liability, as incidences are addressed by states and territories on a case-by-case basis.

While not as extensive as the ANZWQG, the Australian and New Zealand Sediment Quality Guidelines [64] use a similar approach based on weight of evidence. However, the toxicity data that underpin these guidelines require updating, and the entire process needs rejuvenation. Unfortunately, there are no foreseeable updates for the Australian and New Zealand Sediment Quality Guidelines. A range of other guidelines are available that specifically pertain to offshore waste disposal, recreational waters and urban stormwaters. 

#### 2.2.2. Monitoring 

At a national scale, some of the most significant long-term and systematic marine monitoring programs fall under the Integrated Marine Observing System (IMOS) (www.imos.org.au (accessed on 12 November 2022)). The IMOS’s programs capture the open ocean, the continental shelf and coast, and encompass a range of disciplines, including biology, ecology, biochemistry and physics. In addition to programs that track marine mammals and other megafauna such as great white sharks, the IMOS has long-term offshore programs for monitoring natural variations in phytoplankton communities. Berry et al. [65] recently used eDNA metabarcoding of IMOS oceanic samples to illustrate how oceanic communities change over time and respond to pronounced increases in seawater temperature. The IMOS has a direct relationship with Bioplatforms (www.bioplatforms.com (accessed on 13 November 2022)), which invests in infrastructure and supports strategic projects in the ‘omics’ space. For example, Bioplatforms manages a marine microbiome program that investigates microbial communities of seawater, sediment, sponges, seaweeds, corals and seagrasses. However, it should be emphasized that these are not biomonitoring programs in the traditional sense, and rather provide an inventory of the region’s microbial diversity.

While most of the Australian states have routine monitoring programs, in general, these focus on physical and chemical measurements as proxies for ecological conditions; this approach is used because of the costs, latency and impracticalities of obtaining robust ecological data. For example, many estuaries in New South Wales (NSW) are routinely monitored under the NSW Monitoring, Evaluation and Reporting Program (MER) (https://www.environment.nsw.gov.au/topics/water/estuaries/monitoring-and-reporting-estuaries (accessed on 23 October 2022)). One of the most significant programs is southeast Queensland’s Ecosystem Health Monitoring Program (EHMP), which has been running since 2000 (https://hlw.org.au/monitoring-data/ (accessed on 5 July 2022)). This program currently uses physico-chemical data as well as other metrics, such as riparian vegetation and seagrass cover, to assign estuaries a decreasing score from A to F. However, Chariton et al. [59] demonstrated that it would be possible to bolster this program using eDNA metabarcoding to obtain ecological data (benthic eukaryote communities). Furthermore, Graham et al. [66] used the Bayesian risk assessment model to show how the program’s long-term physico-chemical data could be combined with eDNA metabarcoding data to predict the ecological condition of the region’s estuaries. The study also illustrated how a reduction in nutrients from sewage treatment plants may influence the composition of sediment invertebrate communities. While not yet routine at a state-wide scale, there is considerable interest in eDNA approaches as a means to obtain ecological data that is both relatively cost effective and rapid and comprehensively covers biodiversity. Consequently, we anticipate a significant shift towards eDNA routine monitoring programs in the coming years.

### 2.3. China

China is the world’s most populous country and, as well as having a large land area, the country has a sea area of 3 million km^2^. China has jurisdiction over vast continental shelves and exclusive economic zones up to 200 nautical miles off its coasts. China’s rapid economic development and the land-use activities of its coastal cities are increasing the pollution pressures on offshore environments, resulting in serious ecological deterioration. Almost all of China’s coastal seas are under pressure from anthropogenic pollutants, oil spills and multiple marine activities [67]. Thus, the protection of marine coastal zones from anthropogenic stressors is regarded as a matter of great urgency. The main activities in the Chinese coastal marine zones include seawater aquaculture, bathing beaches, coastal tourist resorts, MPAs and disposal of wastes. The following sections summarize the main sources of offshore environmental pollution in China.

The China coastal line is 18,000 km long, and there are over 1500 rivers entering the sea, including the Yalu, Liaohe, Haihe, Yellow, Huaihe, Yangtze and Pearl Rivers. The routine monitoring of pollutants discharged into the sea is managed by the State Oceanic Administration of China (SOA). The current routine coastal environmental monitoring frameworks cover chemical oxygen demand (COD), ammonia nitrogen, nitrate nitrogen, total phosphorus, petroleum, trace metals (including zinc, copper, lead, cadmium and mercury) and arsenic. The average COD value of sewage outlets discharged by the major rivers into the sea has been reported at 1452 × 104 t/a. Except for a brief rise in 2014 and 2015, the COD values of sewage outlets have declined since 2010. It is evident that the Yangtze River, Qiantang River, Pearl River, Minjiang River and Yellow River are the main COD contributors in China. The presence of persistent organic pollutants (POPs) in coastal and estuarine environments has been widely studied by Chinese scientists, including petroleum hydrocarbon, polychlorinated biphenyl (PCBs), organochlorine pesticide (OCPs) and polycyclic aromatic hydrocarbon (PAHs). The average concentrations of the U.S. Environmental Protection Agency’s 16 priority PAHs in surface sediments of Liaodong Bay have shown a downward trend since 2010 [68]. These POPs were all found at Dalian; however, the surface sediment average at the same location was significantly higher in 2016 and 2017, which could be explained by the oil spill from a drilling well [69]. From 2010 to 2017, the average values of ammonia nitrogen and total phosphorus pollutants in the sewage discharged into the sea were 30.9 × 104 and 26.2 × 104 t/a, respectively. The Yangtze, Qiantang, Pearl, Minjiang and Yellow Rivers were the main contributors of ammonia nitrogen and total phosphorus, with maximum levels tested in the sea in 2010 and 2012, respectively. Overall, the total amount of ammonia nitrogen in sewage has shown a downward trend year by year from 2010 to 2017, but the total phosphorus has not decreased. The total amounts of nitrate nitrogen discharged by the major rivers significantly increased from 2010 to 2012, but there are signs the levels have been stabilizing and declining around 2017. Notably, the total amounts of nutrients discharged into the sea via the main rivers increased with the rapid development of the Chinese economy; although, since 2010, there has been an overall downward trend [70]. As one of the most developed economic zones in China, the Yangtze River Delta has been the main source of petroleum pollutants and trace metals into the sea; however, petroleum pollutants have now been reduced following the highest value of 9.5 × 104 t/a recorded in 2012 [69]. Moreover, since 2010, the total amounts of zinc, copper, lead, cadmium and mercury discharged annually into the sea by the main rivers in China have generally decreased, with zinc having the highest proportion of the five trace metals. Arsenic is a globally recognized carcinogen, and the average amount of arsenic discharged into the sea from major Chinese rivers during 2010 and 2017 was 3310 t/a, with the maximum arsenic inflow occurring in 2010 (4226 t/a). Overall, however, the total content of arsenic inflows into the sea has shown a downward trend since 2010, except for brief increases in 2012 and 2014.

The ratio of plastic in floating waste in China has increased in recent years. The data indicate that the average amount of waste on the seashore is significantly higher than the average amount of floating or deposited waste on the seafloor. The average density of seafloor waste was two orders magnitude higher than the average density of seashore and floating waste from 2010 to 2017, and there has been no sign of declining trends. Since 2016, the SOA has organized pilot monitoring programs of marine plastic, and the results showed that the average density of microplastics floating in the surface water of sections of the Bohai Sea, East China Sea and South China Sea was 0.29 particle/m^3^. Floating microplastics mainly included polyethylene, polypropylene and polystyrene, and the main sources were from fishing equipment, debris, film and foam. A 2016 study in the surface waters of the Bohai Sea reported an average microplastic concentration of 0.33 ± 0.34 particle/m^3^ [71].

The development of mariculture can bring economic benefits, but it puts pressure on the environmental quality of surrounding marine environments, bathing beaches and coastal resorts. The quality of seawater and sediment in marine aquaculture areas is monitored annually by the SOA, and the data showed eutrophication in some of the water bodies between 2010 and 2017, including higher contents of inorganic nitrogen and active phosphate. The water quality of recreational bathing beaches was mainly affected by the high content of faecal *Escherichia coli*, the presence of plankton-like algae and the occurrence of red tide. In the estuarine area of Bohai Bay, aquaculture ponds and residential land use have increased, and in the coastal wetlands of the Yangtze River, the area of paddy fields, fishponds and reservoirs accounts for 21%. The land-use type of the Jiulongjiang River Estuary area, which is primarily woodland, is also influenced by urbanization and industrialization. The environmental monitoring of China’s seawater aquaculture covers more than 60 areas, including the Dandong, Sanmen Bay and Nanao aquaculture areas. Collected data have shown that the proportion of aquaculture areas with an ‘excellent’ environmental quality grade for seawater has increased since 2010, with the exception of the period between 2011 and 2012.

#### 2.3.1. Regulations and Guidelines

*The eco-environmental assessment guidance for terrestrial pollution source and near sea area* was issued by the SOA in 2005. This standard mainly focused on sea water and sediment quality in adjacent areas to land-source sewage outlets into the sea. There were only five normative documents in the guidance, and the monitoring items, methods and criteria were not comprehensive. In 2015, the SOA issued a new guidance called *Technical regulations for environmental monitoring and evaluation of land-based sewage outlets and adjacent sea areas*. Sewage discharge, sewage water (salinity, pH, COD, suspended solids, total nitrogen and active phosphate, total phosphorus, petroleum, mercury, cadmium, lead, copper, zinc, chromium, arsenic, total organic carbon, etc.) and sewage biological toxicity were included in the list of monitored items. In addition, the new guidance standard included more parameters, and the number of normative documents was increased to 144. Since 2015, the analytical technology, equipment and monitoring methods have also been continually improved.

In recent years, the National Marine Environmental Monitoring Center of China (NMEMCC) has compiled the Technical Guide for Marine Waste Monitoring, the guideline for marine debris monitoring and assessment and the guideline for marine bathing beach monitoring and assessment. The NMEMCC has selected specific areas to monitor including, for example, the distribution of marine junk annually since 2007. Moreover, until 2017, the total number of areas monitored for marine waste discharge was increased to 49. The monitoring parameters include the current stock and the distribution of surface, beach and seabed garbage, which is consistent with other countries and regional organizations. The floating waste that is monitored includes debris plastics, polystyrene foam fragments, flake wood and plastic wood. The national monitoring stations for sewage outlets into the sea now cover almost the entire coastline of China, with stations located in the Yellow Sea, Bohai Sea, East China Sea and South China Sea.

#### 2.3.2. Monitoring

The NMEMCC (https://www.nmemc.org.cn/ (accessed on 3 January 2022)) coordinates the collection of environmental data to monitor the ecological environment of offshore and coastal areas. The monitoring of key marine fishery zones reached 5.228 million hectares in 2018, according to the *Report on the state of the fishery eco-environment in China*. Seven of the world’s top 10 ports are in China, and they account for 30% of the world’s total container shipping throughput each year. Importantly, the pollution contribution of ports to the coastal environment cannot be overlooked. According to a report released by the Natural Resources Defense Council, environmental impact assessments are an effective tool for managing air and marine pollution caused by the emissions from port-based ships, trucks and operating equipment. For example, the replacement of oil-fired generators by shore-side electric power supply systems could effectively reduce marine pollution associated with fuel consumption. In order to further coordinate the source control of pollution from ships and promote the use of shore power for ships in port, the Ministry of Transport of the People’s Republic of China has issued the *Action of preventing and controlling the pollution from ship and ports (2015–2020).*

Mangroves are the first line of defence for coastal shelterbelt systems; however, in China, areas of mangrove have significantly reduced and now constitute only 2% of the world’s total mangrove forests. "Returning ponds to forests" should be at the forefront of mangrove restoration in China. Since 2001, a series of plans and documents have been issued to protect mangroves, and around 50% of mangroves have been designated as nature reserves. China has now established 38 natural protected areas covering more than 75% of the natural mangrove areas.

Currently, China faces six major environmental challenges: pollution control and emission reduction, improvement of environmental quality, adjustment of regional environmental policies, integrated emission reduction to increase awareness of environmental rights and interests, and effective implementation of policies. More effective real-time monitoring and online data sharing are required. The Civil Code of the People’s Republic of China, introduced on 1 January 2021, has obtained the tort liability for environmental pollution and damage to the ecology, as well as liability for ecological restoration and the scope of compensation for public interest litigation. In the future, China will introduce new laws to monitor and protect coastal zones.

### 2.4. Japan

Japan is an elongated island country consisting of four main islands—Hokkaido (83,424 km^2^), Honshu (231,127 km^2^), Shikoku (18,789 km^2^) and Kyushu (42,231 km^2^)—and the Okinawa islands (2281 km^2^). The total area of Japan is approximately 378,000 km^2^ and, as the climate is relatively mild/temperate and humid, there are many species of plants and animals inhabiting both terrestrial and aquatic ecosystems in subarctic to subtropical zones. These natural characteristics of Japan have supported agriculture, forestry, fisheries and tourism nationwide for many years. Particularly after the Meiji era (1868–), industrialization in Japan progressed rapidly and markedly, driven by the influence of Western European countries. Notably, Japan was enormously impacted by the outbreak of World War II (1941–1945), and following the Korean War (1950–1953), the country faced special demands and a period of high economic growth (1955–1973). As a result of the country’s financial recovery and extended growth, Japan became the second largest economy in the world. However, the economic progress led to rapid changes in land use and expansions of industrial areas, such as the coastal developments of Keihin/Keiyo (i.e., Tokyo, Kanagawa and Chiba), Chukyo (i.e., Nagoya), Hanshin (i.e., Osaka and Kobe) and Kitakyushu (i.e., northern Fukuoka). This phase also witnessed the destruction of nature in Japan and consequently resulted in occurrences of severe environmental pollution. For example, the reclamation of inshore/coastal waters has caused water pollution such as eutrophication and red tide events. Notably, mass propagation of phytoplankton has been linked to changes in the material cycle of nitrogen and phosphate, etc. Occurrences of hypoxia have also been observed in enclosed bays and adjacent areas.

Water pollution has resulted in the decline in total catch and changes in species composition in Japanese commercial fisheries (for example, [72]). Other factors identified as contributing to the decline of target commercial fisheries species (see below) include overfishing and the loss/destruction of tidal flats and shallow waters, which are critical habitat and nursery areas for aquatic animals. It is difficult to estimate the impacts of pollution (namely, quantitative decrease and qualitative deterioration) on the value of local coastal zones in relation to tourism; the development of coastal zones has resulted in economic growth through increasing tourist numbers. Pollution by trace metals and harmful chemical substances is prevalent because of insufficient treatment of industrial waste effluents. Japan has also experienced major impacts from environmental pollution, including Minamata disease (the first victim was formally identified in 1956), which has occurred in several areas of Japan (for example, [73]). 

Located on the east-central coast of Japan, Tokyo Bay is semi-enclosed with a surface area of 960 km^2^ and an average depth of 15 m. Tokyo Bay is a site of high cultural and economic value; originally a productive area, commercial fisheries peaked at approximately 140,000 metric tons in 1960, but the total catch has since steadily decreased [72]. The Tokyo Bay ecosystem is affected by a range of anthropogenic pressures from the surrounding metropolitan areas. These pressures include eutrophication, commercial and recreational fishing, input of chemical contaminants and modification of habitat by coastal development [72]. However, fisheries statistics do not reveal changes in stock sizes of species that are not targets for the fishing industry. Data related to the stock-size variations of several taxa are useful as fundamental sources of information for assessing shifts in the megabenthic community structure as well as for understanding the mechanisms behind the variations. Fisheries-independent surveys of the biota, including both target and non-target species, are necessary to confirm temporal trends in stock sizes and species composition in the community [74].

#### 2.4.1. Regulations and Guidelines

The Air Pollution Control and Water Pollution Prevention Acts were established in 1968 and 1970 to protect human and environmental health. The Environmental Agency of Japan was established in 1971 following the 1970 debate on national diet. Since 1979, the Environmental Agency of Japan has developed water use regulations to control the input of nitrogen and phosphate into inshore/coastal waters. Consequently, levels of COD and concentrations of total nitrogen and total phosphate have decreased since the 1980s. However, the area of hypoxic water mass and the duration of hypoxia occurrences are yet to be improved and continue to decline (Round-table Conference on Measures for Enclosed Coastal Sea, Japan (2007) [75]). This situation suggests that controlling the input of nitrogen and phosphate into inshore/coastal waters does not necessarily result in a decrease in hypoxic water mass in terms of area and duration. To address the issue of hypoxic water mass, it is important to consider frameworks such as the Integrated Coastal Area and River Basin Management (ICARM) to maintain and improve the sustainability of the coastal ecosystem services [76].

Environmental standards were established for the protection of human health, air, noise, water and soil environments. For coastal waters, there are environmental quality standards for COD, total nitrogen and total phosphate. An environmental quality standard for bottom dissolved oxygen (DO) has been set to resolve issues related to hypoxia in enclosed bays and adjacent areas [77].

The Act on the Evaluation of Chemical Substances and Regulation of their Manufacture was established in 1973 to protect human and environment health, and the Act has been revised over time. When first established, the intention was to protect human health under the joint jurisdiction of the Ministry of International Trade and Industry (MITI) and the Ministry of Health and Welfare (MHW). During this early period, the Environmental Agency of Japan could only express an opinion to both the MITI and MHW. In the revision of 2003, the Act incorporated the evaluation of ecological effects by chemical substances. Over time, Japan has introduced several acts and laws for environmental protection, including the Agricultural Chemicals Regulation Law, which was established in 1948. The Basic Environment Law was introduced in 1993, along with the main environmental law. The Basic Environment Law provides the fundamental directions for the protection and conservation of the environment. The Environmental Agency of Japan was re-organized into the Ministry of the Environment in January 2001 following a restructure of the central government.

#### 2.4.2. Monitoring

The Environmental Health and Safety Division of the Environmental Agency of Japan has been publishing annual reports on “Chemicals in the Environment” since 1974. These surveys provide analytical data on chemicals that are likely to pose environmental risks, such as POPs. Tributyltin (TBT) and triphenyltin (TPhT) are significant contaminants in Japan. It is well known that imposex (i.e., a superimposition of male genital tracts, such as the penis and vas deferens, in females) is induced and developed by certain organotin compounds used in antifouling paints, including TBT and TPhT [78]. Monitoring surveys to elucidate population-level effects related to reproductive failure involving imposex or masculinization of females were conducted in *Thais clavigera*, the ivory shell *Babylonia japonica* and the giant abalone *Haliotis madaka*. From the 1990s to the 2000s, these surveys were conducted at sites in Japan with contamination by TBT and TPhT. The surveys used combined methods of histological examination of gonadal tissue and chemical analysis to identify TBT, TPhT and their metabolites in tissue [79,80].

As a result of the 2011 Tohoku earthquake (Mw 9.0) and the tsunami on 11 March 2011, three nuclear reactors at the Fukushima Daiichi Nuclear Power Plant (FDNPP), owned by the Tokyo Electric Power Company (TEPCO), went into meltdown. Hydrogen explosions in the reactor buildings resulted in the emission of hundreds of petabecquerels (PBq) of radionuclides into the environment [81]. The amount of radionuclide leakage from the FDNPP was about one-tenth the amount released by the 1986 Chernobyl Nuclear Power Plant disaster in Ukraine, where the total release of radionuclides was estimated to be 5300 PBq, excluding noble gases [81]. The severity of the nuclear accident at the FDNPP has raised concerns about the contamination of aquatic organisms by radionuclides in both freshwater and marine environments. By the end of March 2011, the Japanese government had begun to determine activity concentrations of radionuclides (i.e., gamma emitters) in fishery products (i.e., fishes, crustaceans, molluscs and echinoderms) for radioprotection purposes. In general, the contamination of marine organisms by radio caesium is higher in demersal fish than in pelagic fish; specimens of both demersal and pelagic fish collected off Fukushima Prefecture have shown higher radionuclide activity contamination levels than those collected off other prefectures. The activity concentrations of radio caesium in fish tissue, however, have decreased since the FDNPP accident in most fish sampled from the region (e.g. Wada et al., [82]). Fishing operations in the Fukushima Prefecture region were resumed on a trial basis in June 2012 and, since then, the areas and species targeted have been gradually recovering [83].

To date, environmental monitoring surveys have focused on chemical residues in select biota. More information on temporal trends in environmental factors (i.e., contaminant levels and environmental criteria) and population/community levels are required to better formulate and implement measures to ensure environmental safety.

## 3. Future Trends

The Asia-Pacific coastal regions face a range of anthropogenic threats, especially around heavily populated areas such coastal China [15]. It is well-recognized that healthy oceans and thriving coastal and freshwater ecosystems are essential for economic growth and food production [84]. Priorities for the protection and improvement of the health of coastal ecosystems, natural stock and customary practices have been highlighted through initiatives such as the establishment of MPAs, which provide a useful pathway to achieve sustainable blue economies. The cumulative impacts of multiple stressors cannot be predicted from single stressor studies, as there may be different interactions when examining, for example, community distributions and resilience [85]. The review has highlighted some similarities and differences between the countries that have been summarized in Table 1. As expected, some of the stressors are the same, including impacts from urbanization and agricultural practices. China and Japan, having much larger populations, have many more agencies involved in the management of coastal zones when compared to Australia and New Zealand. It is worth noting that the New Zealand RMA uses an ‘effects-based’ assessment approach based on the concepts of sustainable management, with the integrated management of resources incorporating community and indigenous (Māori) values which is quite unique.

Climate change is of growing concern and will continue to have profound impacts on ecosystems, and thus the interactions with other stressors need to be carefully managed [86]. Restoration investment costs to address the degradation of marine coastal habitats will continue to increase as restoration practices improve and become larger in scale [87]. The consequences of human-induced climate change must also be considered, as they may result in increasing extinction rates or ecosystem collapses in coastal zones [88], which particularly impact estuarine environments [89]. The changes in species distribution and spread in response to climate change are an important cause of population declines and extinctions [90]. These pressures on ecosystems will increasingly affect human populations through a range of challenges such as food security. This is particularly important to communities in the Pacific Islands region that are acutely affected by the decline of small-scale fisheries under climate change [91]. 

The protection of important marine ecosystem services requires coordinated national and regional legislation to minimize the potential adverse impacts of anthropogenic activities [92]. The European Union MSFD directive is based on 11 qualitative descriptors, including GES descriptor 8 on chemical contaminants, which includes the objective, “Concentrations of contaminants are at levels not giving rise to pollution effects” [19]. There is no coordinated monitoring framework currently in place in the Asia-Pacific region. The EBM framework was developed to provide services by maintaining or enhancing ecological structure and function [93]. EBM considers the need for some coordination across countries that share ocean ecosystems to overcome the limitations of political or administrative boundaries. However, there are multiple potential challenges concerning the implementation of EBM, including the lack of integration between agencies and departments and inadequate policy alignment; there are also a variety of other socio-political factors to overcome in the North Pacific initiative [93]. 

Framework options will have to be fit for purpose for the Asia-Pacific region and integrate the specific and unique aspects and values encountered in the South Pacific. Such frameworks will require data collection over a range of spatial scales (from point measurements to continuous measurements, e.g., sea surface temperature, sea-level height and chlorophyll a from remotely sensed satellite data) within the respective countries/jurisdictions. In addition to internationally coordinated efforts, there is a need to recognize local strategies. For instance, the New Song Noumea Strategy represents an important shift away from failed centralized models of coastal fisheries management toward support for community-based management of marine resources [91]. The integration of traditional knowledge and management practices in broader integrated coastal planning is essential. A series of demonstration sites and pilot activities were identified as having contributed to the development of best practice in coastal habitat and land-based pollution management in the South China Sea; these initiatives used new approaches such as incorporating traditional indigenous knowledge [23].

Future research should ensure that the chemical analysis of environmental specimens is combined with biological or ecological analysis at population/community levels. This approach will provide more information about temporal trends in environmental factors (i.e., pollutants or contaminants) and variations of specific populations or communities inhabiting a certain area; these variations may be possible effects or biological response(s) induced by temporal changes in the local environmental factor(s). Examples of such research in Japan include monitoring surveys on imposex caused by organotins in gastropod molluscs, monitoring surveys on the megabenthic community and environment in Tokyo Bay, and monitoring surveys on invertebrate and megabenthic communities in intertidal zones and coastal waters off Fukushima [78,94,95,96].

## 4. Solutions

This review is part of the Special Issue, “Ecotoxicity of Contaminants in Water and Sediment”, that highlights some of the key challenges facing contaminant risk assessors and managers globally [97]. We have highlighted several of the significant challenges that need to be addressed to develop approaches that better monitor and manage the health of marine environments in the Asia-Pacific region. Effective solutions will require the implementation of regionally coordinated monitoring programs. The development of ecosystem-based management approaches needs to include assessments of cumulative effects that are underpinned by collaborative engagement with the wider community; this will ensure the inclusion of local values and traditional knowledge. 

First, an overarching issue relates to the development of coordinated monitoring frameworks accompanied by standardized methods that facilitate rigorous regional and global comparative analyses. Essentially, this will involve the systematic collection and sharing of long-term data using standardized protocols. An effective integrated framework will require data collection over a range of spatial scales, from point measurements to continuous measurements within the respective jurisdictions. The overall effectiveness of both a coordinated monitoring framework and policy instruments for the Asia-Pacific region will depend on strong leadership and commitment as well as regional ownership and cooperation. Currently, several existing coordinated networks have been implemented, including the Global Ocean Observing System (GOOS), which aims to provide oceanographic data for long-term predictions of weather and climate under three critical themes of climate, operational services and marine ecosystem health [98]. The South Pacific Regional Environment Programme (SPREP) (currently the SPC Secretariat of the Pacific Community) has also implemented action plans that include monitoring and assessments of the state of the environment in the region. However, the development of a more comprehensive and shared monitoring framework for the Pacific represents a future priority. Such standardized monitoring initiatives would enable large-scale processes to be compared nationally as well as internationally to facilitate improved monitoring of marine ecosystem health and early detection of climate change over time. Unique considerations in the Asia-Pacific region include sea-level monitoring assessments for the large number of raised coralline islands and low-lying atolls dispersed over the expanse of the southwestern Pacific Ocean, a region that is particularly vulnerable to sea-level rise.

In concert with coordinated monitoring frameworks, the development of ecosystem-based approaches to management would also provide benefits to the Asia-Pacific region. Given the scope and magnitude of environmental challenges facing natural resource management, there is increasing demand for more holistic, ecosystem-based management approaches [44]. Ecosystem-based management (EBM) includes the comprehensive integrated management of all human activities based on the best available scientific knowledge about the ecosystem and its dynamics. EBM is also a place-based approach that aims to balance long-term ecosystem health and functioning, which collectively provide the essential ecosystem services for local populations [99]. Marine spatial planning (MSP) is one process that offers a promising opportunity for more integrated management [99]. MSP identifies which areas of the ocean are appropriate for different uses or activities in order to reduce conflicts and achieve ecological, economic and social objectives. Within the Asia-Pacific region, there are many opportunities for growing the blue economy in the form of deep-sea minerals, pharmaceuticals and alternative renewable energy sources. Given the likelihood of further growth of the blue economy, sustainable development should be supported through monitoring frameworks that are embedded within an EBM framework. However, this will also require careful consideration of monitoring frameworks that can detect cumulative impacts across multiple sectors. Cumulative effects, through additional new marine industries, climate change and other stressors, can reduce environmental resilience and increase the risk of environmental or economic collapse. The increasing potential for unexpected and irreversible environmental shifts emphasizes the need to adopt a precautionary approach and manage the exploitation of marine resources well within the known environmental limits. In addition to establishing widespread monitoring and reporting programs in the Asia-Pacific region, a key challenge remains: there is a significant need to enhance knowledge about the conditions under which sudden, disruptive and substantive undesirable changes are likely to occur, and researchers and policy-makers must understand the potential implications of such changes [100]. 

Finally, local and traditional ecological knowledge (LTK) is increasingly recognized as an important component of scientific research, conservation and resource management [101,102]. Local knowledge can be a significant source of information, especially where there is limited environmental data or gaps in the scientific literature, which is of particular relevance for the Asia-Pacific marine environment. For example, LTK is being used to provide historical and contemporary baseline information, guide stewardship techniques, improve conservation planning and resolve management disputes [101]. To be effective, LTK needs to be genuine, trust-building and involve wider communities from the early stages of research; this approach will ensure a shared interest in project objectives, including the visions and expected outcomes of seascapes [101,103]. In the Asia-Pacific region, environmental management and monitoring have been enhanced when decision-making and planning implementation are developed using collaborative processes with indigenous knowledge, local communities and resource users [44]. In general, collaborative forms of planning and decision-making can resolve conflicts over scarce resources and provide opportunities for building social capital and trust [104]. While participatory approaches can be successful, improved outcomes also rely on a well-designed process [105]. Some of these principles include prioritizing long-term solutions over short-term benefits, collective engagement of all key stakeholders, delivering a negotiated consensus on sustainability goals, and providing guidance through a skilled facilitator [105]. Examples of collaborative work in the Asia-Pacific region have already demonstrated that social-ecological systems can shift toward more sustainable trajectories using well-designed participatory processes that support collaborative planning [44,105].

## Figures and Tables

**Table 1 toxics-11-00277-t001:** Comparisons between the 4 countries within the Pacific region. The shared key stressors facing coastal zones in these regions include sediment, nutrient enrichment, chemical and waste pollution, plastic, pathogens, invasive species that are all compounded by climate change.

Region	Population living by the coast	Pollution sources	Main governance bodies	Policies
New Zealand	>75 % of 5 million live within 10 km of the coast ^1^	Urban sources Farming Forestry	Maori	Resource Management Act (1991)
			Regional Council	
			Environment Protection Authority (EPA) Ministry for the Environment (MfE) Maritime NZ	
Australia	85% of 22 million live within 50 km of the coast	Urban sources Farming Mining	Environmental Protection Authority (EPA)	Environment Protection and Biodiversity Conservation Act 1999
			Department of Climate Change, Energy, the Environment and Water	
China	45% of 1.4 billion live in coastal provinces ^2^	Fishery Transport Tourism Discharges	Ministry of Agriculture and Rural Affairs Ministry of Transport Maritime Safety Administration Ministry of Ecology and Environment	Environmental Protection Act 2017 Action of Preventing and Controlling the Pollution from Ship and Ports (2015–2020) Guideline for marine bathing beach monitoring and assessment; Guideline for marine debris monitoring; Assessment; Technical guide for Marine waste monitoring; Technical regulations for environmental monitoring and evaluation of land-based sewage outlets and adjacent sea areas
Japan	85% of 124.6 million live in coastal areas ^3^	Industrial waste Transport Farming Domestic waste	Ministry of Economy, Trade and Industry, Ministry of Health, Labour and Welfare, Ministry of the Environment Ministry of Land, Infrastructure, Transport and Tourism, Ministry of the Environment Ministry of Agriculture, Forestry and Fisheries, Ministry of the Environment Ministry of Health, Labour and Welfare, Ministry of the Environment	Act on the Evaluation of Chemical Substances and Regulation of Their Manufacture, Air Pollution Control Act, Water Pollution Prevention Act Air Pollution Control Act Agricultural Chemicals Regulation Law, Water Pollution Prevention Act Water Pollution Prevention Act, Waste Disposal Law, The Basic Environment Law

^1^ Sourced from the Parliamentary Commissioner for the Environment (PCE), Managing our estuaries report (2020): https://pce.parliament.nz/media/jp0h0r3l/report-managing-our-estuaries-pdf-44mb.pdf (accessed on 9 February 2023). ^2^ Source from the Nation Bureau of Statistics, Bulletion of the seventh National Census: http://www.gov.cn/guoqing/2021-05/13/content_5606149.htm (accessed on 9 February 2023). ^3^ Statistics Bureau of Japan https://www.stat.go.jp/english/index.html (accessed on 9 February 2023).

## Data Availability

Not applicable.

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
