# Peer review of "Monitoring the Health of Coastal Environments in the Pacific Region—A Review"

_toxics, 2023, doi:10.3390/toxics11030277_

Round 1

Reviewer 1 Report

The article submitted for review under the title: "Monitoring the health of coastal environments in the Pacific region-a review" is an interesting scientific work dealing with the broadly understood anthropogenic impact on the marine environment. The described region of the world is perceived as one of the most polluted in the world. Collecting scientific information on this subject is extremely valuable. All chapters are very extensive. For each of the described regions there is a subchapter on monitoring and Regulations and guidelines. I don't know if the article shouldn't have a different structure. Wouldn't it be better to describe the same areas in general, but focus on common or divergent aspects of environmental protection, especially those two subchapters "Monitoring" and "Regulations and guidelines". The final chapters of "Future trends" "Solutions" are interesting and sufficient. In my opinion, the article is interesting and raises important aspects, but it requires major revision. The authors should restructure the article in such a way that the monitoring and legislative aspects are combined into one subchapter in which the similarities or differences between the countries described: China, Australia, Japan and New Zealand will be described.

Author Response

Reply: The authors thank you for these very useful comments. We are glad that you are viewing this review as important as we agree that the range of pressures in this region is detrimental to the health of unique and highly valuable coastal zones. The co-authors have debated at length how to structure such a challenging topic over large and varied geographical area with a variety of environments and management approaches. In fact, this was the key justification behind undertaking this review. We wanted to provide a description of the types of stressors in these regions all the way to the policy in place to manage the risks and protect the respective receiving environments. We structured used the same headings for all 4 regions. We agree with the reviewer that it does make it challenging to the reader to compare regions and highlight differences. To address this, we have included a summary table (Table 1) that gives a snapshot of the 4 regions under key parameters. We have also provided additional text to define key differences, e.g.:

The review has highlighted some similarities and differences between the countries that have been summarized in Table 1. As expected, some of the stressors are the same including impacts from urbanization and agricultural practices. China and Japan having much larger populations, have many more agencies involved in the management of coastal zones when compared to Australia and New Zealand. It is worth noting that the New Zealand RMA uses an 'effects-based' assessment approach based on the concepts of sustainable management, the integrated management of resources incorporating community and indigenous (Māori) values which is quite unique.”

Reviewer 2 Report

CommeSuggestions for Authors

(will be shown to authors)

Comments and suggestions for authors were presented in the following file. 

Author Response

Reply: Thank you for this comment that the manuscript needed to be reviewed. We appreciate your concern, and the manuscript has now been reviewed by an English editor (L Fisher, as par the acknowledgements section). The edits were quite extensive so we have included a version of the paper with track changes to highlight the changes. The manuscript has now a much better flow.  

Round 2

Reviewer 1 Report

I thank the authors for considering my comments. In its current form, the article meets the requirements to be published in the journal Toxics.